# Monocarboxylate transporter dependent mechanism is involved in proliferation, migration, and invasion of human glioblastoma cell lines via activation of PI3K/Akt signaling pathway

Chen Gao [ID]*[◉], Binni Yang[◉], Yurong Li, Wenjuan Pei

Department of General Practice, The 940th Hospital of Joint Logistics Support Force of Chinese People's Liberation Army, Lanzhou, China

◉ These authors contributed equally to this work.
* gc2006418@163.com

## Abstract

Glioblastoma multiforme is one of the most common primary tumors of the central nervous system, with a very poor prognosis. Cancer cells have been observed to upregulate pH regulators, such as monocarboxylate transporters (MCTs), with an increase in MCT4 expression being observed in several malignancies. MCT4/ recombinant cluster of differentiation 147 (CD147) transporter complex was reported to stimulate vascular endothelial growth factor (VEGF) via the phosphatidylinositol 3 kinase (PI3K) /protein kinase B (Akt) pathway, which has been proven to mediate glioblastoma invasion and migration. The present study aimed to clarify the role of the MCT4/CD147 transporter complex in glioblastoma cell proliferation, migration, and invasion. In this work, lentiviral vectors were used to overexpress MCT4/CD147 and small interfering RNA (siRNA) was used to silence MCT4/CD147 in the human glioma cell lines U87 and U251, respectively. The effects on cell proliferation, migration and invasiveness, as well as the protein expression levels of MCT4 and CD147, extracellular lactate content and Akt activation were assessed by MTT, wound-healing and invasion assays, western blotting and colorimetric method, respectively. The analysis results suggested that cell proliferation, migration, invasion, and Akt activation were decreased by siRNA in all cell lines, but were increased by lentivirus-mediated MCT4 overexpression. These findings suggest that inhibiting the activity and expression of the MCT4/CD147 transporter complex via metabolic-targeting drugs, particularly in cells with a high rate of glycolysis, should be explored as a novel strategy for glioblastoma treatment.

## Introduction

Despite advances in treatment, the prognosis for patients with glioblastoma (WHO grade IV) remains poor. Cancer cells exhibit uncontrolled cell proliferation, migration and invasion as a

**Data Availability Statement:** Raw images of Western blotting assays have been uploaded as Supporting Information files, and other datasets

used and/or analyzed during the current study cannot be shared publicly due to restrictions on the disclosure of research data of military units by the Ethics Committee. Data are available from the official website (https://www.gsinfo.net.cn/index.php/Institution/ywbmptzx.html) for researchers who meet the criteria for access to confidential data

**Funding:** This work was supported by the Natural Science Foundation of Gansu Province, China (grant number 23JRRA1668) and the Foundation of 940th Hospital Research Project, Lanzhou, Gansu Province, China (grant number 2023YXKY037). The funders had no role in study design, data collection and analysis, decision to publish, or preparation of the manuscript.

**Competing interests:** The authors have declared that no competing interests exist.

result of a switch in energy metabolism from oxidative phosphorylation to aerobic glycolysis [1], even in the presence of oxygen, known as the Warburg effect [2]. Tumor cells efflux lactate into the extracellular microenvironment, preventing intracellular acidosis and consequent cell death, while the acidic extracellular environment enables migration and invasion via extracellular matrix degradation, and it also promotes vascularization and immune suppression. and the more-acidic extracellular microenvironment and more-alkaline intracellular milieu are considered hallmarks of cancer [3]. To achieve this, tumor cells upregulate pH regulators such as monocarboxylate transporters (MCTs) [4]. MCTs belong to the SLC16 gene family of 14 members, of which MCT1-4 are proton symporters that mediate the transmembrane transport of lactate, pyruvate and ketone bodies [5]. MCT4 is one of the many target genes of hypoxia-inducible factor 1α (HIF-1α) and is expressed almost exclusively in astrocytes, although increased expression has been reported in several malignancies [6, 7]. Lactate produced by glycolytic tumor cells plays an important role in the tumor microenvironment and is associated with poor prognosis [8]. MCT4 and the co-expressed chaperone protein CD147/basigin [9] are overexpressed in the plasma membrane of glioblastoma cells compared to levels in diffuse astrocytomas and non-neoplastic brain [10]. MCT4 requires association with recombinant cluster of differentiation 147 (CD147) in the endoplasmic reticulum for trafficking to the plasma membrane [11, 12]. In addition, MCT4 and Basigin/CD147 have been reported to stimulate vascular endothelial growth factor (VEGF) through the phosphatidylinositol 3 kinase (PI3K) /protein kinase B (Akt) signaling pathway [13], which has been shown to mediate glioblastoma invasion and migration. Given the pivotal role of MCT4 in maintaining glycolytic metabolism in tumor cells, a better understanding of the role of the MCT4/CD147 transporter complex in mediating glioblastoma cell invasion, migration, and proliferation, and of the role of Akt signaling are essential. In the present study, we used lentivirus-mediated gene transfer and small interfering RNA (siRNA) to overexpress or silence MCT4/CD147, respectively, in two human glioma cell lines. We then explored the effect of these genetic modifications on the proliferation, migration, and aggressiveness of the cell lines by MTT, wound-healing, and invasion assays, respectively, and subsequently evaluated MCT4 and CD147 expression and Akt activation.

## Materials and methods

### Cell lines and culture

The high-grade human glioma cell lines U87 and U251 were maintained in Dulbecco's Modified Eagle's Medium (DMEM) supplemented with 10% fetal bovine serum (FBS) and 1% penicillin–streptomycin solution at 37°C, in a 5% $CO_2$/95% air (balanced nitrogen and 85% humidity) incubator (i.e., normoxic conditions), and routinely passaged at 2–3 day intervals.

### Plasmid production by lentiviral transfer and infection

Human MCT4 cDNA (Open Biosystems; Thermo Scientific, Huntsville, AL, USA) was cloned into a lentiviral vector pLentilox (pLL) 3.7 to generate the pLL3.7-MCT4 vector. A pLL3.7 vector only expressing green fluorescent protein (GFP) was used as the empty vector control. pLL3.7-GFP or pLL3.7-MCT4 was co-transfected with psPAX2 vector and envelope protein vector p-CMV-VSV-G into 293FT packaging cells using Effectene Transfection Reagent (Qiagen, Valencia, CA, USA). Forty-eight hours after transfection, cell supernatants containing virus were harvested and used to infect cell lines. The cell lines were replated into six-well culture plates at $1\times10^5$ cells/well and transduced with lentiviruses containing pLL3.7-GFP (GFP) or pLL3.7-MCT4 (MCT4 overexpression) by centrifugation at $920 \times g$ for 1 h at 25°C.

## siRNA-mediated silencing of MCT4 or CD147

The validated siRNA for human MCT4-specific siRNA (ID, 107345) and a negative-control siRNA (ID, AM4611)were purchased from Ambion (Austin, TX, USA). The siRNAs were transfected with Lipofectamine 2000 transfection reagent (#11668–027, Invitrogen, Carlsbad, CA) in U87 and U251cell lines according to the protocol of the manufacturer. The mixture was transferred to an electroporation cuvette with a 2-mm gap (Eppendorf, Hamburg, Germany) and nucleofection was performed in the Nucleofector device using the U-031 program.

CD147 expression in MCT4 overexpressing cells was also silenced with siRNA (ID: 215973, Ambion) using the same experimental protocol as above, with the negative control (ID, AM4611) siRNA as a non-silencing control.

## Real-time polymerase chain reaction

MCT4 and CD147 mRNA levels in cell lines were detected by RT-PCR, according to the manufacturer's protocol. Quantitative RT-PCR was performed for 45 cycles at 95°C for 15 s and 60°C for 1 min using an IQ5 Multicolor real-time PCR detection system (Biorad, München, Germany) for a 1/10 volume of the RT reaction mixture and TaqMan Gene Expression Assays (Applied Biosystems, Foster City, CA, USA). USA), with human glyceraldehyde phosphate dehydrogenase (GAPDH) as a loading control. The primer sequences were as follows: GAPDH (forward, *5'-GCACCGTCAAGGCTGAGAAC-3'*; reverse, *5'-TGGTGAAGACGCC AGTGG-3'*, assay ID Hs03929097_g1); MCT4 (forward, *5'-TGCGGCCCTACTCTGTCT-3'*; reverse, *5'-TCTTCCCGATGCAGAAGAAG-3'*, assay ID Hs00358829_m1), and CD147 (forward, *5'-TGTAAAACGACGGCCAGT-3'*; reverse, *5'-CAGGAAACAGCTATGACC-3'*, assay ID Hs00616498_CE) Relative gene expression levels were calculated using iQ5 Optical System Software and repeated at least three times.

## Western blot analysis

Proteins extracted from glioma cell lines were incubated with primary antibodies against MCT4 (catalog #: PAB21410, 1:500; Abnova Corporation, Taipei City, Taiwan), CD147 (catalog #: 23187, 1:1000; Cell Signaling Technology, Inc., Danvers, MA, USA), phospho-Akt (Ser473) (catalog #: 9271, 1:1000; Cell Signaling Technology, Inc.), and total-Akt (catalog #: 9272, 1:1000; Cell Signaling Technology, Inc.), with α-tubulin (catalog #: 2144, 1:1000; Cell Signaling Technology, Inc.) as a loading control. Signal intensity was analyzed using Image J (version 1.38x, National Institutes of Health, Bethesda, MD, USA) in the positive digital photographs of the membranes per formula: Relative optic density = (Target band density)/ (α-tubulin densiy). Representative WB images are presented in the figures.

## Immunofluorescence

Cells were incubated with rabbit polyclonal anti-MCT4 (catalog #: PAB21410, 1:2000; Abnova Corporation) or rabbit polyclonal anti-CD147 (catalog #: PA5-29787, 1:1000; Invitrogen Life Technologies, Rockford, IL, USA), and mouse monoclonal GFAP (catalog #: 3670, 1:500; Cell Signaling Technology, Inc.) antibodies. Fluorescein isothiocyanate-conjugated anti-mouse and tetramethyl rhodamine isothiocyanate-conjugated anti-rabbit (both 1:100; CoWin Biotech, Co. Ltd., Beijing, China) were used as secondary antibodies. Images were analyzed using Image J software.

## Determination of extracellular lactate content

After successful lentiviral transfection or siRNA silencing interference of the cell lines, extracellular lactate contents of cell lines were determined by using the L-Lactate Assay Kit

(Cayman Chemical, Bath: 0521418, USA).the culture medium was taken for lactate content determination by colorimetric method [3, 14]. The operation was performed in strict accordance with the instructions of the kit.

## MTT assay

Cells from each group ($5\times10^3$) were seeded in 96-well plates and incubated for 7 days. MTT assay was performed every 24 h, according to the manufacturer's protocol. Color intensity was measured at 490 nm using an enzyme linked immunosorbent assay plate reader (Tecan Sunrise Remote, Queensland, Austria).

## Wound-healing assay

Cells were cultured in medium with 10% FBS and mitomycin C (200μg/ml). After 1 h, three homogenous artificial wounds were created in the monolayer with a sterile plastic 100-μl micropipette tip. Cells that migrated into the wounded area or cells with extended protrusions at the border of the wound were visualized and photographed under a light microscope (×200) 24 h after wounding. Relative migration distances were analyzed using Image J Software.

## Transwell invasion assay

Cell culture chambers (24 wells) with Transwell inserts (Corning Life Sciences, Corning, NY, USA) with an 8-μm pore membrane precoated with 50μl Matrigel (BD Biosciences, San Jose, CA, USA) were used in the Transwell invasion assay. Cells in each group were plated at a density of $1\times10^4$ per upper well in 200 μl DMEM without FBS, and the lower chamber was filled with 500 μl of DMEM plus 10% FBS. Cultures were incubated at 37˚C and cells were allowed to invade for 24 h. Gently but firmly remove Matrigel from the inside of the transwell insert by use of a cotton tipped applicator to ensure all Matrigel and remaining non-migratory cells are removed. Cells on the lower surface of the filter were fixed for 20 min in absolute ethyl alcohol and stained with crystal violet after brief air-drying. The mean number of migrated and invaded cells was counted in eight preselected microscopic fields under a light microscope (×200) and quantified with ImageJ software.

## Statistical analysis

All statistical analyses were carried out using SPSS v.16.0 for Windows (SPSS Inc., Chicago, IL, USA). All quantitative values are presented as mean ± standard deviation (SD), values conformed to normal distribution, and differences were assessed by one-way analysis of variance (ANOVA) with Bonferroni or Tamhane's post-hoc tests. *P* values < 0.05 were considered statistically significant.

# Results

## Verification of gene-transduction efficiency

At 48 h after gene transduction, U87 and U251cells respectively were divided into five groups: a blank group (normal cultured cells without lentiviral transduction); an MCT4 lentiviral group (transduced with pLL3.7-MCT4 lentiviral vector); an empty vector group (transduced with pLL3.7-GFP lentiviral vector); an MCT4 siRNA group (transduced with MCT4-specific siRNA); and a control siRNA group (transduced with negative control siRNA). MCT4 protein and mRNA levels in the cells were assessed by western blot, immunofluorescence, and real-time PCR. MCT4 protein was visible as a 43-kDa band in the immunoblot (Fig 1A). The results of relative quantitative analysis are shown in Fig 1B. MCT4 expression levels in U87

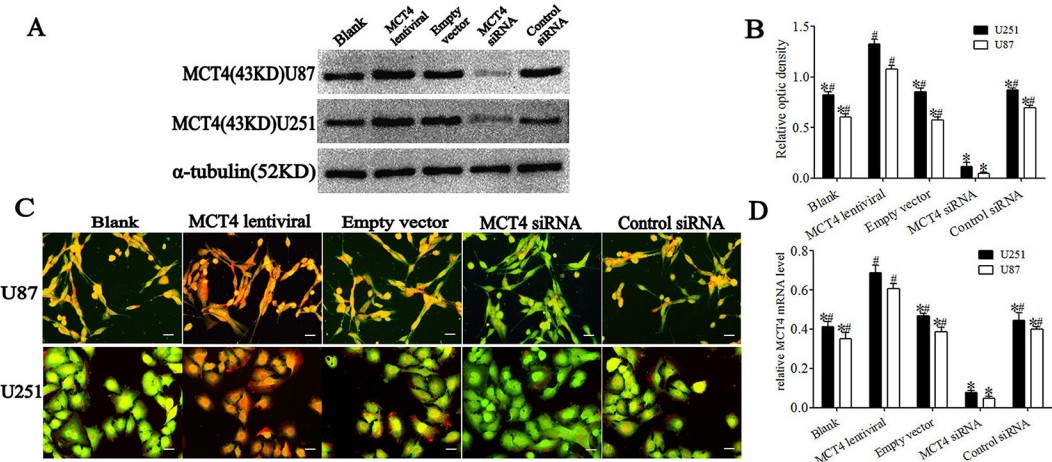

**Fig 1. MCT4 protein and gene expression levels in U87 and U251 cell lines after lentivirus transduction or siRNA interference. A.** Representative immunoblot showing MCT4 protein levels corresponding to different protocols. **B.** Results of relative quantitative analysis by immunoblotting. **C.** Expression of MCT4 (red) and the glial marker GFAP (green) were determined by immunofluorescence double labeling. Scale bar = 50 μm. **D.** Relative quantitative analysis of MCT4 mRNA levels by real-time PCR. *$P < 0.05$ vs. MCT4 lentiviral group; #$P < 0.05$ vs. MCT4 siRNA group (ANOVA with post hoc Tamhane correction; n = 3 independent experiments per group).

and U251 cells were 80.1% and 63.7% higher ($P < 0.05$), respectively, in the MCT4 lentiviral group compared with the empty vector group, while MCT4-specific siRNA inhibited protein expression by 94.3% and 86.0% ($P < 0.05$), respectively, in the MCT4 siRNA group compared with the control siRNA group. However, there were no significant differences among the blank, empty vector, and control siRNA groups ($P > 0.05$). The immunofluorescence results were consistent with the trend observed for immunoblotting (Fig 1C), while GFAP expression levels were the same in all groups. The trends in MCT4 mRNA expression were identical to those for MCT4 protein (Fig 1D). MCT4-specific siRNA inhibited MCT4 mRNA expression in U87 and U251 cell lines by 88.2% and 82.5% ($P < 0.05$), respectively, compared with the control siRNA group, while lentiviral plasmid transfer significantly increased MCT4 mRNA expression by 70.8% and 65.1% ($P < 0.05$), respectively.

## Extracellular lactate content of cell lines (Table 1)

The extracellular lactate content of MCT4 lentiviral group of U87 and U251 cell lines was 0.375±0.023 mmol/g and 0.268±0.011 mmol/g, respectively, which was significantly higher

**Table 1. Extracellular lactate content of cell lines in each group.**

| Group | Average extracellular lactate content | |
| --- | --- | --- |
| | **U87** | **U251** |
| Blank | 0.163±0.010*# | 0.101±0.009* |
| MCT4 lentiviral | 0.375±0.023# | 0.268±0.011# |
| Empty vector | 0.119±0.012*# | 0.125±0.013*# |
| MCT4 siRNA | 0.047±0.015* | 0.072±0.025* |
| Control siRNA | 0.127±0.016*# | 0.112±0.029* |

Data are indicated in mmol/g as mean ± SD.

*$P < 0.05$ vs. MCT4 lentiviral group; #$P < 0.05$ vs. MCT4 siRNA group (ANOVA with post hoc Tamhane correction; n = 3 independent experiments per group)

than that in the other groups ($P < 0.05$). MCT4 siRNA group had extracellular lactate content of 0.047±0.015 mmol/g and 0.072±0.025 mmol/g, respectively, significantly lower than the other groups($P < 0.05$).

## In vitro effect of CD147 silencing on MCT4-overexpressing cell lines

To determine if MCT4 expression required association with CD147, we silenced CD147 expression in MCT4-overexpressing cell lines by siRNA. MCT4-overexpressing cells transfected with negative control siRNA were used as a non-silencing control. Western blot analysis revealed decreased expression of both glycosylated and non-glycosylated CD147, as well as MCT4, in the MCT4 lentiviral+CD147 siRNA group compared with the MCT4 lentiviral group and MCT4 lentiviral +control siRNA group (Fig 2A). The results of relative quantitative analysis are shown in Fig 2B ($P < 0.05$). We observed loss of CD147 expression in the MCT4 lentiviral+CD147 siRNA group (Fig 2C), using fluorescence microscopy. CD147 siRNA efficiently silenced CD147 mRNA ($P < 0.05$), whereas no change was observed in MCT4 mRNA levels ($P > 0.05$) compared with the MCT4 lentiviral group or MCT4 lentiviral +control siRNA group (Fig 2D).

## Identification of MCT4/CD147-associated cell proliferation, migration and invasion in U87 and U251 cell lines

We assessed the role of the MCT4/CD147 transporter complex in cell proliferation, migration and invasion via MTT assay, wound-healing assay and transwell invasion assay, respectively. Compared with blank cells, cell proliferation curves constructed from independent experiments revealed decreased cell proliferation in MCT4 or CD147 siRNA interference cells, but increased cell proliferation in lentivirus-transduced cells ($P < 0.05$) (Fig 3A and 3B).

Cell-migration distances of U215 and U87 cells were significantly reduced in the MCT4 siRNA (26.86±6.32 µm, 42.35±7.68 µm, respectively) and MCT4 lentivirus+CD147 siRNA groups (31.25±7.12 µm, 46.45±8.02 µm, respectively), but enhanced in the MCT4 lentivirus group (82.52±7.45 µm, 96.56±5.21 µm, respectively) at 24 h compared with the blank group ($P < 0.05$) (Fig 3C and 3D).

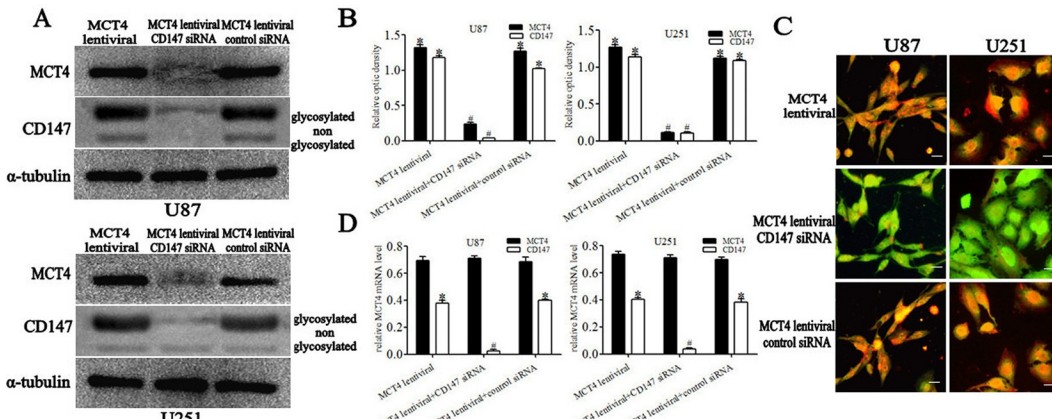

**Fig 2. *In vitro* effect of CD147 silencing on MCT4-overexpressing cell lines. A.** Representative immunoblot showing MCT4 and CD147 protein levels corresponding to different protocols. **B.** Results of relative quantitative analysis by immunoblotting. **C.** Expression of CD147 (red) and the glial marker GFAP (green) were determined by immunofluorescence double labeling. Scale bar = 25 µm. **D.** Relative quantitative analysis of MCT4 and CD147 mRNA levels by real-time PCR. *$P < 0.05$ vs. MCT4 lentiviral+CD147 siRNA group; #$P < 0.05$ vs. MCT4 lentiviral group (ANOVA with post hoc Bonferroni correction; n = 3 independent experiments per group).

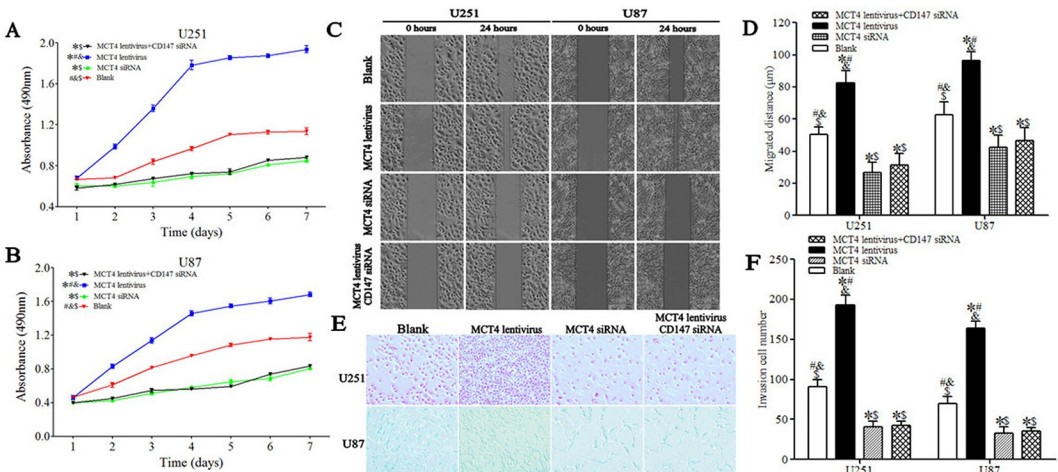

**Fig 3. MCT4/CD147 transporter complex-associated cell proliferation, migration and invasion in U87 and U251 cell lines. A.** Cell proliferation curves of U251 cells. **B.** Cell proliferation curves of U87 cells. The curves constructed from independent experiments revealed decreased cell proliferation in MCT4 and CD147 siRNA interference cells, but increased cell proliferation in lentivirus-transduced cells. **C.** Representative images of migration assay at 0 and 24 h (100× magnification). **D.** Distance of migrated cells in each group. Results expressed as mean±SD. Cell migration in U215 and U87 cells was significantly inhibited in MCT4 and CD147 siRNA interference cells compared with blank cells, but enhanced in lentivirus-transduced cells at 24 h. **E.** Representative images of invading cells (200× magnification). **F.** Number of invading cells in each group. Results expressed as mean±SD. U251 and U87 cell invasiveness was significantly attenuated following MCT4 or CD147 interference compared with blank cells, but enhanced in lentivirus-transduced cells. $*P < 0.05$ versus blank group, $^{\#}P < 0.05$ versus MCT4 siRNA group, $^{\&}P < 0.05$ versus MCT4 lentivirus +CD147 siRNA group, $^{\$}P < 0.05$ versus MCT4 lentivirus group (ANOVA with post hoc Tamhane correction; n = 3 per cell lines and per day).

Compared with blank cells, downregulation of MCT4 or CD147 significantly reduced the invasiveness of siRNA interference cells, while lentivirus-mediated MCT4 overexpression resulted in increased cell invasion in lentivirus-transduced cells ($P < 0.05$) (Fig 3E and 3F).

## PI3K/Akt signaling pathway is related to MCT4/CD147 complex in glioma cell lines

We assessed the signal transduction molecules phospho-Akt and total-Akt by immunoblotting to investigate the relationship between MCT4/CD147 and the PI3K/Akt signaling pathway. Compared with the blank group, phospho-Akt levels were increased by 68.9% in U87 cells in the MCT4 lentiviral group, with more significant reductions of 70.0% and 67.8% in the MCT4 siRNA and MCT4 lentiviral+CD147 siRNA groups ($P < 0.05$), respectively. A similar tendency was observed in U251 cells. Phospho-Akt expression was upregulated by 69.1% in the MCT4 lentiviral group and downregulated by 49.2% and 43.5% in the MCT4 siRNA and MCT4 lentiviral+CD147 siRNA groups ($P < 0.05$), respectively. However, there was no significant difference between the MCT4 siRNA and MCT4 lentiviral+CD147 siRNA groups ($P > 0.05$). In addition, there was no obvious change in total-Akt expression in glioma cell lines ($P > 0.05$) (Fig 4A and 4B).

## Discussion

A high rate of glycolysis is a metabolic hallmark of cancer. Indeed, some tumor cells, especially in advanced cancers, may exhibit aerobic glycolysis, a metabolic phenotype known as the Warburg effect [15]. Elevated levels of lactate were correlated with poor patient prognosis in high-grade gliomas [16]. Increasing evidence indicates that lactate provides a fuel for oxidative metabolism in oxygenated tumor cells [17], acts as a signaling agent in tumor and endothelial

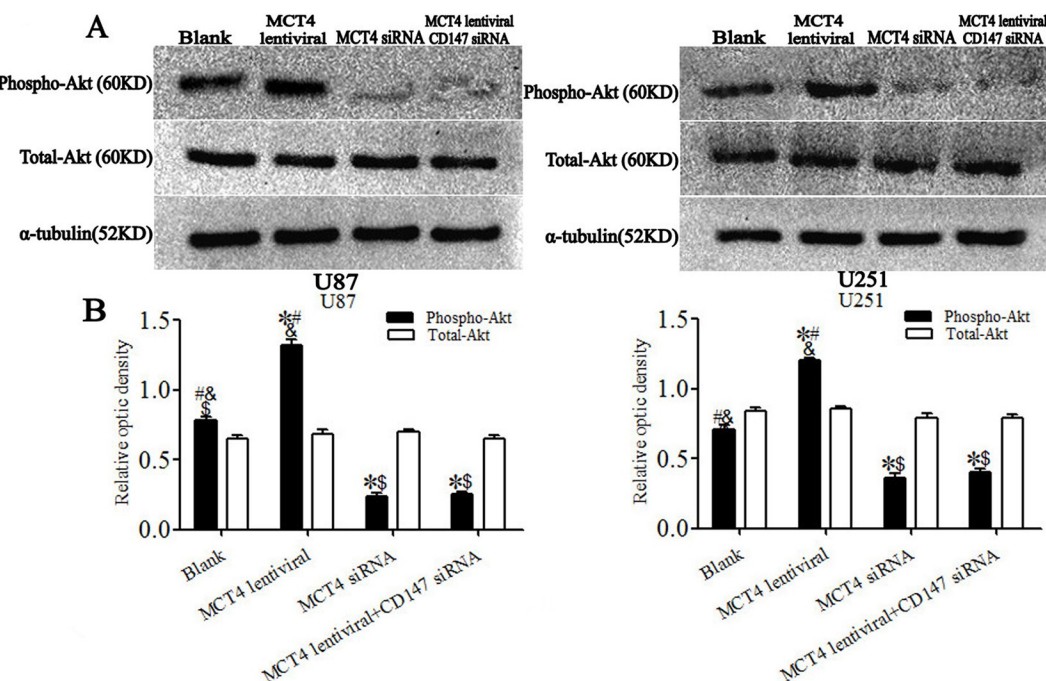

**Fig 4. PI3K/Akt-related signaling proteins after MCT4/CD147 lentivirus transduction or siRNA interference in U87 and U251 cell lines. A.** Representative immunoblot showing phospho-Akt and total-Akt protein levels corresponding to different protocols. **B.** Results of relative quantitative analysis by immunoblotting. Levels of phospho-Akt were reduced in MCT4 and CD147 siRNA interference cells, but significantly increased in lentivirus-transduced cells compared with blank cells, while total-Akt expression was unchanged. *$P < 0.05$ versus blank group, #$P < 0.05$ versus MCT4 siRNA group, &$P < 0.05$ versus MCT4 lentivirus +CD147 siRNA group, $$P < 0.05$ versus MCT4 lentivirus group (ANOVA with post hoc Bonferroni correction; n = 3 independent experiments per group).

cells [18], and is an important contributor to wound repair and angiogenesis [19]. MCTs have been confirmed as prominent facilitators of lactate exchanges between cancer cells with different metabolic behaviors, and between cancer and stromal cells. We therefore addressed the function and regulation of MCTs as novel anticancer targets. Glioblastomas are very aggressive human neoplasms, presenting high resistance to currently available therapies [20]. Glycolysis is upregulated in glioblastomas, accompanied by an increased ratio of lactate to pyruvate [21]. MCT4 largely associated with the export of lactate in cells with high rates of glycolysis related to hypoxic energy production [22]. We therefore explored the potential value of MCT4 as a molecular target in glioblastoma therapy.

Recent studies identified MCT4 as a heteromeric transporter composed of a catalytic a-subunit (MCT) and an accessory h-subunit (CD147),which was identified as an extracellular matrix metalloproteinase inducer in cancer cells [23], and subsequently found to be expressed at high levels in metastatic tumor cells [24]. Previous studies showed that MCT4 siRNA reduced the level of MCT4 but not CD147 mRNA in MDA-MB-231 cells, while immunoblot analysis confirmed loss of MCT4 and glycosylated CD147 protein expression in cells treated with MCT4 siRNA. In the present study, silencing CD147 resulted in reduced MCT4 and CD147 protein expression levels, but not in MCT4 mRNA. siRNA-mediated silencing of MCT4 or CD147 had the same effect on proliferation, migration, and invasion of glioma cell lines. These results suggest that the function of the MCT4/CD147 transporter complex depends on the integrity of the molecular constructs.

Gomes et al. found that overexpression of MCT4 was associated with a poor prognosis in prostate cancer [25]. Gallagher et al. showed that silencing of MCT4 resulted in decreased cancer cell migration *in vitro*, by mechanisms involving interaction between MCT4 and integrin β1 [26]. Kubelt et al. indicate the existence of individual differences in the regional distribution of MCT1 and MCT4 and suggest that both transporters have distinct connections to glioblastoma multiforme progression processes, which could contribute to the drug resistance of MCT-inhibitors [27]. In the present study, MCT4 silencing by siRNA interference resulted in significant decreases in cell proliferation, migration, and invasion in glioma cell lines, while these were increased in lentivirus-mediated MCT4 overexpressing cell lines. The aforementioned results are in accordance with those of other recent studies, which demonstrated that classical MCT inhibitors and interference of MCTs reduced the migration and invasion capacities of breast, lung, and glioma cells [28, 29]. These results suggest that the inhibition of lactate transport and consequent arrest of glycolytic flux increase the sensitivity of glioma cells to standard therapies. MCTs are thus promising molecular targets for adjuvant therapy in patients with glioblastomas, sensitizing glioma cells to standard therapies whilst having minimal impact on the integrity and viability of normal brain. In addition, previous studies showed that CHC had an inhibitory effect on tumor-associated vascularization. Lactate increases production of VEGF, the major angiogenic factor in the microenvironment [30]. The MCT4/CD147 complex was reported to stimulate VEGF production through the PI3K/Akt signaling pathway [13]. This pathway is an important intracellular signaling pathway regulating the proliferation, migration, and invasion of glioblastomas, and overactive in many cancers, thus reducing apoptosis and allowing proliferation. However, the pathway is also necessary for promoting the growth and proliferation, as opposed to differentiation, of adult stem cells, and specifically neural stem cells. Researchers are currently trying to determine the appropriate balance between proliferation and differentiation, and to exploit this balance in the development of various therapies [31]. The present study demonstrated that the PI3K/Akt signaling pathway was activated in MCT4-overexpressing glioma cell lines, but inhibited in MCT4 or CD147 siRNA interference cells. We hypothesized that MCT4/CD147 transporter complex mediated the biological characteristics of glioma cells in an Akt-dependent manner, and downregulation of MCT4/CD147 by siRNA interference decreased tumor size and the number of blood vessels, likely because of impaired tumor glycolytic metabolism and decreased VEGF production, mediated by decreased microenvironmental lactate concentrations.

## Conclusion

In conclusion, we suggested that inhibiting the MCT4/CD147 transporter complex via metabolic-targeting drugs, particularly in cells with high rates of glycolysis, could be explored as a potential new strategy for glioblastoma treatment.

## Supporting information

**S1 Raw images.** Westernblot showing the entire gel for Fig 1A (MCT4 U87 and U251, α-tubulin), Fig 2A (MCT4 U87 and U251, CD147 U87 and U251, α-tubulin), Fig 4A (Phospho-Akt U87 and U251, Total-Akt U87 and U251, α-tubulin) were adapted from this western blot. (ZIP)

## Acknowledgments

The authors would like to express sincere gratitude to M.M. Hui Li for providing the cell lines and M.M. Jinhua Wang for her contributions to this work.

## Author Contributions

**Formal analysis:** Binni Yang.

**Funding acquisition:** Yurong Li.

**Writing – original draft:** Chen Gao.

**Writing – review & editing:** Wenjuan Pei.

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
