## [Decision Letter · Decision Letter 0]

23 Jun 2024

PONE-D-24-13072Monocarboxylate transporter dependent mechanism is involved in proliferation, migration, and invasion of human glioblastoma cell lines via activation of PI3K/Akt signalingpathwayPLOS ONE

Dear Dr. Gao,

Thank you for submitting your manuscript to PLOS ONE. After careful consideration, we feel that it has merit but does not fully meet PLOS ONE’s publication criteria as it currently stands. Therefore, we invite you to submit a revised version of the manuscript that addresses the points raised during the review process.

Reviewer 1: Major points - The scientific background in the introduction section is very poor. More information is needed to give a clear overview of the current state of knowledge and the novelty of the results reported in the manuscript. - How do you explain the very small increase of MCT4 in the “overexpressing cells”, considering that it is almost the same in all control cell lines (blank(?), control siRNA, empty vector)? - Based on the biochemical role of the MCT4 transporter, the correct characterization of the cell line models generated here requires the metabolic profiles of these cells (e.g. intracellular pH, lactate levels, ECAR/OCR levels, ATP production). - Lines 48-50: please, clarify this statement. - Considering that MCT4 regulates the transport of lactate across the membrane and modulates the mitochondrial activity of cells, the MTT assay is completely inappropriate to test cell viability. To detect the effects of MCT4 expression on cell proliferation and cell cycle distribution, more appropriate assays such as the trypan blue assay, cytofluorimetry and the BrdU/EdU assay are required. - Based on the previous findings discussed in the manuscript, measurement of VEGF production may be useful. - To demonstrate the effect of the MCT4/CD147 axis on the PI3K/Akt signaling pathway, a simple analysis of Akt phosphorylation is not sufficient. Based on previous findings (Zhou Y, 2020; Bu X, 2021), the authors should improve the results to substantiate their findings. The levels of phospho-GSK3b/GSK3b and Nrf2 should be checked. - Considering that both Akt and ATM/ATR inhibitors are available (and there are no selective MCT4 inhibitors to date, Goldberg WF, 2023), assessing the differential sensitivity of the described cancer cell models to these agents could be of value for a translational impact of the results presented here. - The reference list is insufficient. Minor points - Line 180: What do you mean by “no drug treatment”? - Blank: is an invalid control definition, please replace it in the text, legend and figures - Results of relative quantitative analysis by immunoblotting: please explain better how you performed it - Figure 4: Quantification of Akt phosphorylation should be expressed as the ratio of P-Akt band density/Akt band density relative to actin level. - Methods: There is not enough information to reproduce some of the experiments. Please provide more details for Western blot and immunofluorescence analysis. Add a detailed description of the invasion assay where the removal of the Matrigel is not mentioned. - Paragraph 3.3: Explain better the results of migration and invasion. Give the values as a percentage reduction of the wound compared to the corresponding control. - Remove the comments from all figure legends and explain the technical details of the experiments shown (what they represent and how the values given in the graphs were calculated). - Please provide the originals of the immunoblot analyses. 

Reviewer 2: In this manuscript authors clarify the role of the pH regulator monocarboxylate transporter MCT4 and its accessory h-subunit CD147 in glioblastoma cells. Briefly, they demonstrated that inhibiting the activity and expression of the MCT4/CD147 transporter complex decreases cell proliferation, migration, invasion, and Akt activation, while lentivirus-mediated MCT4 overexpression reverts the system in glioblastoma cells. Manuscript needs some minor and major revisions: - An extensive English editing is required. I suggest contacting a native speaker. - There are several typing errors in the whole text. Please, check them in the whole text. - Genes must be written in italic. - The acronyms in the abstract should be replaced with the relative extended. - Introduction is too short and assume several concepts that could be unknown for the readers. To this, I suggest improving the background section. - Densitometry of MCT4(43KD)U251 in “Control siRNA” sample (Fig.1B) appears not to correspond to the band intensity of the relative WB (Fig.1A). Please check it. - The quality of Western Blots analyses is poor. All Wb are oversaturated. Please replace them with blots with reduced exposure, when it’s possible, and add the original blots as supplementary materials. - To test the invasion ability of glioblastoma cells, authors have performed the transwell assay. However, the 3D spheroid invasion assay may represent the most relevant assay for this goal, particularly for glioblastoma cell lines, as well as to better mimic tumor behavior in vivo [doi: 10.3791/52686]. Briefly, 3D tumor spheroids are embedded into 3D ECM. It is expected to see that non-invasive cancer cells stay as compact spheroids with a distinct border to the surrounding ECM and do not show any obvious signs of invasion. On the contrary, invasive cells start to invade into the surrounding matrix and display outgrowth from the spheroids [doi: 10.1371/journal.pone.0293475; doi: 10.3389/fcell.2023.1272667]. - What is the difference between glycosylated and non-glycosylated CD147 (Fig. 2A)? Please explain this concept in more depth. - In the Discussion section, authors assume the involvement of PI3K/Akt signaling pathway. However, they have performed only the WB analysis of Akt protein expression without verify the involvement of PI3K or other key players in this pathway. I highly recommend exploring the other components or activation of downstream molecules.

We look forward to receiving your revised manuscript.

Kind regards,

Nicola Amodio, PhD

Academic Editor

PLOS ONE

Journal Requirements:

2. Thank you for stating the following financial disclosure: "This work was supported by the Natural Science Foundation of Gansu Province, China (grant number 23JRRA1668) and the Foundation of 940th Hospital Research Project, Lanzhou, Gansu Province, China (grant number 2023YXKY037)."

3. We note that your Data Availability Statement is currently as follows: "All relevant data are within the manuscript and its Supporting Information files."

Additional Editor Comments:

Dear Dr. Gao,

two reviewers have assessed your manuscript.

Although findings look interesting, both reviewers are asking major revisions to enhance the quality of the manuscript.

Reviewer 1:

Major points

- The scientific background in the introduction section is very poor. More information is needed to give a clear overview of the current state of knowledge and the novelty of the results reported in the manuscript.

- How do you explain the very small increase of MCT4 in the “overexpressing cells”, considering that it is almost the same in all control cell lines (blank(?), control siRNA, empty vector)?

- Based on the biochemical role of the MCT4 transporter, the correct characterization of the cell line models generated here requires the metabolic profiles of these cells (e.g. intracellular pH, lactate levels, ECAR/OCR levels, ATP production).

- Lines 48-50: please, clarify this statement.

- Considering that MCT4 regulates the transport of lactate across the membrane and modulates the mitochondrial activity of cells, the MTT assay is completely inappropriate to test cell viability. To detect the effects of MCT4 expression on cell proliferation and cell cycle distribution, more appropriate assays such as the trypan blue assay, cytofluorimetry and the BrdU/EdU assay are required.

- Based on the previous findings discussed in the manuscript, measurement of VEGF production may be useful.

- To demonstrate the effect of the MCT4/CD147 axis on the PI3K/Akt signaling pathway, a simple analysis of Akt phosphorylation is not sufficient. Based on previous findings (Zhou Y, 2020; Bu X, 2021), the authors should improve the results to substantiate their findings. The levels of phospho-GSK3b/GSK3b and Nrf2 should be checked.

- Considering that both Akt and ATM/ATR inhibitors are available (and there are no selective MCT4 inhibitors to date, Goldberg WF, 2023), assessing the differential sensitivity of the described cancer cell models to these agents could be of value for a translational impact of the results presented here.

- The reference list is insufficient.

Minor points

- Line 180: What do you mean by “no drug treatment”?

- Blank: is an invalid control definition, please replace it in the text, legend and figures

- Results of relative quantitative analysis by immunoblotting: please explain better how you performed it

- Figure 4: Quantification of Akt phosphorylation should be expressed as the ratio of P-Akt band density/Akt band density relative to actin level.

- Methods: There is not enough information to reproduce some of the experiments. Please provide more details for Western blot and immunofluorescence analysis. Add a detailed description of the invasion assay where the removal of the Matrigel is not mentioned.

- Paragraph 3.3: Explain better the results of migration and invasion. Give the values as a percentage reduction of the wound compared to the corresponding control.

- Remove the comments from all figure legends and explain the technical details of the experiments shown (what they represent and how the values given in the graphs were calculated).

- Please provide the originals of the immunoblot analyses.

Reviewer 2:

In this manuscript authors clarify the role of the pH regulator monocarboxylate transporter MCT4 and its accessory h-subunit CD147 in glioblastoma cells. Briefly, they demonstrated that inhibiting the activity and expression of the MCT4/CD147 transporter complex decreases cell proliferation, migration, invasion, and Akt activation, while lentivirus-mediated MCT4 overexpression reverts the system in glioblastoma cells.

Manuscript needs some minor and major revisions:

- An extensive English editing is required. I suggest contacting a native speaker.

- There are several typing errors in the whole text. Please, check them in the whole text.

- Genes must be written in italic.

- The acronyms in the abstract should be replaced with the relative extended.

- Introduction is too short and assume several concepts that could be unknown for the readers. To this, I suggest improving the background section.

- Densitometry of MCT4(43KD)U251 in “Control siRNA” sample (Fig.1B) appears not to correspond to the band intensity of the relative WB (Fig.1A). Please check it.

- The quality of Western Blots analyses is poor. All Wb are oversaturated. Please replace them with blots with reduced exposure, when it’s possible, and add the original blots as supplementary materials.

- To test the invasion ability of glioblastoma cells, authors have performed the transwell assay. However, the 3D spheroid invasion assay may represent the most relevant assay for this goal, particularly for glioblastoma cell lines, as well as to better mimic tumor behavior in vivo [doi: 10.3791/52686]. Briefly, 3D tumor spheroids are embedded into 3D ECM. It is expected to see that non-invasive cancer cells stay as compact spheroids with a distinct border to the surrounding ECM and do not show any obvious signs of invasion. On the contrary, invasive cells start to invade into the surrounding matrix and display outgrowth from the spheroids [doi: 10.1371/journal.pone.0293475; doi: 10.3389/fcell.2023.1272667].

- What is the difference between glycosylated and non-glycosylated CD147 (Fig. 2A)? Please explain this concept in more depth.

- In the Discussion section, authors assume the involvement of PI3K/Akt signaling pathway. However, they have performed only the WB analysis of Akt protein expression without verify the involvement of PI3K or other key players in this pathway. I highly recommend exploring the other components or activation of downstream molecules.

Reviewers' comments:

Reviewer's Responses to Questions

**Comments to the Author**

1. Is the manuscript technically sound, and do the data support the conclusions?

Reviewer #1: No

Reviewer #2: Partly

2. Has the statistical analysis been performed appropriately and rigorously? 

Reviewer #1: Yes

Reviewer #2: Yes

3. Have the authors made all data underlying the findings in their manuscript fully available?

Reviewer #1: No

Reviewer #2: Yes

4. Is the manuscript presented in an intelligible fashion and written in standard English?

Reviewer #1: Yes

Reviewer #2: Yes

5. Review Comments to the Author

Reviewer #1: In this manuscript, the authors describe the role of the MCT4 transporter in glioblastoma. They hypothesize that the MCT4/CD147 complex promotes proliferation, migration and invasion by activating the Akt signaling pathway. The rationale for the experiments performed is quite good, but in my opinion some points should be addressed to improve the quality of the manuscript. The author's statement in the discussion in lines 344-351 can hardly be supported by the manuscript in its present form.

Major points

- The scientific background in the introduction section is very poor. More information is needed to give a clear overview of the current state of knowledge and the novelty of the results reported in the manuscript.

- How do you explain the very small increase of MCT4 in the “overexpressing cells”, considering that it is almost the same in all control cell lines (blank(?), control siRNA, empty vector)?

- Based on the biochemical role of the MCT4 transporter, the correct characterization of the cell line models generated here requires the metabolic profiles of these cells (e.g. intracellular pH, lactate levels, ECAR/OCR levels, ATP production).

- Lines 48-50: please, clarify this statement.

- Considering that MCT4 regulates the transport of lactate across the membrane and modulates the mitochondrial activity of cells, the MTT assay is completely inappropriate to test cell viability. To detect the effects of MCT4 expression on cell proliferation and cell cycle distribution, more appropriate assays such as the trypan blue assay, cytofluorimetry and the BrdU/EdU assay are required.

- Based on the previous findings discussed in the manuscript, measurement of VEGF production may be useful.

- To demonstrate the effect of the MCT4/CD147 axis on the PI3K/Akt signaling pathway, a simple analysis of Akt phosphorylation is not sufficient. Based on previous findings (Zhou Y, 2020; Bu X, 2021), the authors should improve the results to substantiate their findings. The levels of phospho-GSK3b/GSK3b and Nrf2 should be checked.

- Considering that both Akt and ATM/ATR inhibitors are available (and there are no selective MCT4 inhibitors to date, Goldberg WF, 2023), assessing the differential sensitivity of the described cancer cell models to these agents could be of value for a translational impact of the results presented here.

- The reference list is insufficient.

Minor points

- Line 180: What do you mean by “no drug treatment”?

- Blank: is an invalid control definition, please replace it in the text, legend and figures

- Results of relative quantitative analysis by immunoblotting: please explain better how you performed it

- Figure 4: Quantification of Akt phosphorylation should be expressed as the ratio of P-Akt band density/Akt band density relative to actin level.

- Methods: There is not enough information to reproduce some of the experiments. Please provide more details for Western blot and immunofluorescence analysis. Add a detailed description of the invasion assay where the removal of the Matrigel is not mentioned.

- Paragraph 3.3: Explain better the results of migration and invasion. Give the values as a percentage reduction of the wound compared to the corresponding control.

- Remove the comments from all figure legends and explain the technical details of the experiments shown (what they represent and how the values given in the graphs were calculated).

- Please provide the originals of the immunoblot analyses.

Reviewer #2: In this manuscript authors clarify the role of the pH regulator monocarboxylate transporter MCT4 and its accessory h-subunit CD147 in glioblastoma cells. Briefly, they demonstrated that inhibiting the activity and expression of the MCT4/CD147 transporter complex decreases cell proliferation, migration, invasion, and Akt activation, while lentivirus-mediated MCT4 overexpression reverts the system in glioblastoma cells.

Manuscript needs some minor and major revisions:

- An extensive English editing is required. I suggest contacting a native speaker.

- There are several typing errors in the whole text. Please, check them in the whole text.

- Genes must be written in italic.

- The acronyms in the abstract should be replaced with the relative extended.

- Introduction is too short and assume several concepts that could be unknown for the readers. To this, I suggest improving the background section.

- Densitometry of MCT4(43KD)U251 in “Control siRNA” sample (Fig.1B) appears not to correspond to the band intensity of the relative WB (Fig.1A). Please check it.

- The quality of Western Blots analyses is poor. All Wb are oversaturated. Please replace them with blots with reduced exposure, when it’s possible, and add the original blots as supplementary materials.

- To test the invasion ability of glioblastoma cells, authors have performed the transwell assay. However, the 3D spheroid invasion assay may represent the most relevant assay for this goal, particularly for glioblastoma cell lines, as well as to better mimic tumor behavior in vivo [doi: 10.3791/52686]. Briefly, 3D tumor spheroids are embedded into 3D ECM. It is expected to see that non-invasive cancer cells stay as compact spheroids with a distinct border to the surrounding ECM and do not show any obvious signs of invasion. On the contrary, invasive cells start to invade into the surrounding matrix and display outgrowth from the spheroids [doi: 10.1371/journal.pone.0293475; doi: 10.3389/fcell.2023.1272667].

- What is the difference between glycosylated and non-glycosylated CD147 (Fig. 2A)? Please explain this concept in more depth.

- In the Discussion section, authors assume the involvement of PI3K/Akt signaling pathway. However, they have performed only the WB analysis of Akt protein expression without verify the involvement of PI3K or other key players in this pathway. I highly recommend exploring the other components or activation of downstream molecules.

6. PLOS authors have the option to publish the peer review history of their article (what does this mean?). If published, this will include your full peer review and any attached files.

Reviewer #1: No

Reviewer #2: No

---

## [Author Response · Author response to Decision Letter 0]

27 Aug 2024

(1) Reviewer 1 pointed out that the statement in the discussion in lines 344-351 can hardly be supported by the manuscript in its present form. As the author of the manuscript, I personally agree with this view. This section is the concluding part of this study. Because changes in molecules downstream of the PIK3/Akt signalling pathway were not further tested, the MCT4/CD147 transporter complex mediated the biological characteristics of glioma cells in an Akt-dependent maner is speculative, and further testing of VEGF and related downstream molecules is needed to provide more adequate support in further studies.

(2) According to the comments of reviewers 1 and 2, the scientific background of the introductory section has been enriched, the novelty of this study has been presented, and relevant important concepts have been referenced. The corresponding bibliography of manuscript references has been extensively revised.

(3) Following the comments of reviewer 1, the following explanation is given regarding the very low increase of MCT4 in the lentiviral group. Results both section 3.1 and Figure 1 clearly showed that the increase of MCT4 was obvious after lentivirus transduction of cell lines, which showed statistical difference compared to other groups. The results of Western blotting and immunofluorescence detection of protein level and mRNA detection of gene level can confirm each other.

(4) Reviewer 1 pointed out that further determination of lactate levels to corroborate the biological function of MCT4 is a very professional suggestion. To this end, extracellular lactate content was additionally measured, and the trend of change was generally consistent with the change in MCT4 expression. Relevant content has been added to the Materials and Methods and Discussion sections.

(5) Clarification of lines 48-50 of the manuscript as follows: MCT4 and Basigin/CD147 have been reported to stimulate vascular endothelial growth factor (VEGF) through the phosphatidylinositol 3-kinase/Akt (PI3K/Akt) signalling pathway [12], which has been shown to mediate glioblastoma invasion and migration. This formulation has been confirmed in relevant studies[Tang Y, Nakada MT, Rafferty P, Laraio J, McCabe FL, Millar H, et, al. Regulation of vascular endothelial growth factor expression by EMMPRIN via the PI3Kakt signaling pathway. Mol Cancer Res. 2006; 4:371-377.]

(6) Cytofluorimetry is a technique for analysing cell properties by measuring fluorescent signals in cells or tissues. It is mainly used to study the physiological and pathological states of cells, including cell proliferation, apoptosis, and drug response. BrdU/EdU assays are two commonly used methods in cell proliferation assays, they track cell proliferation by labelling newly synthesised DNA, and are the most accurate method for detecting cell proliferation. Cell proliferation was also detected by MTT and Tepan blue cell assay, which are similar methods for the initial assessment of cell proliferation capacity with simplicity. In this study, MTT assay was chosen for the initial detection of tumour cell line proliferation, and for subsequent in-depth studies, the reviewer-suggested method must be chosen to accurately determine at the genetic level.

(7) This is clearly stated in both the abstract and introduction sections of the manuscript, MCT4 and Basigin/CD147 have been reported to stimulate vascular endothelial growth factor (VEGF) through the phosphatidylinositol 3-kinase/Akt (PI3K/Akt) signalling pathway. Therefore, I strongly agree with the reviewer's comment to include the measurement of VEGF in further in-depth studies so that the findings can be more fully substantiated.

(8) Thanks for the constructive comments from reviewer 1 and 2. GSK-3β is a multifunctional serine/threonine kinase belonging to the glycogen synthase kinase subfamily, which is a negative regulator of glucose homeostasis and is involved in energy metabolism, inflammation, endoplasmic reticulum stress, mitochondrial dysfunction, and apoptotic pathways. GSK-3β is a key downstream protein of the PI3K/Akt signalling pathway, and Akt can phosphorylate Ser9 of GSK-3β and inhibit its enzymatic activity. So Akt plays an important role in energy metabolism. In addition, Nrf2 is a key factor in cellular regulation of oxidative stress, and is the most sensitive signal for scavenging excessive intracellular ROS to combat oxidative stress. It is involved in various cellular activities, including maintenance of redox balance, proliferation, metabolism and apoptosis. It has been shown that the PI3K/AKT signalling pathway can regulate the activity of Nrf2, and the phosphorylation of AKT can further help Nrf2 to achieve the nuclear-plasma shuttling, which can lead to the enhancement of endogenous antioxidant in cells. The present study initially investigated the effect of monocarboxylic acid transporter proteins on the Akt signalling pathway. Thus, the detection of molecules downstream of the Akt signalling pathway, whether GSK-3β or Nrf2, could provide even stronger support for the manuscript's conclusions.

(9) In the present study, the expression of MCTs was modulated by lentiviral transfection and small RNA interference, and the corresponding changes in the Akt signalling pathway were initially investigated without using inhibitors. It is advisable to use sensitive inhibitors for further investigation of the downstream signalling pathways of the Akt signalling pathway.

(10) Line 180: The phrase ‘no drug treatment’ means no lentiviral treatment, i.e. a normal cultured tumour cell line is used as a blank control. Because of ambiguity, it has been changed to ‘without lentiviral transduction’.

(11) Paragraph 2.10 of revised manuscript: Add a detailed description of the transwell invasion assay in the manuscript as blow: Gently but firmly remove Matrigel from the inside of the transwell insert by use of a cotton tipped applicator to ensure all Matrigel and remaining non-migratory cells are removed.

(12) Paragraph 3.4 of revised manuscript: Comparison of the migration distances of the groups of cells in the manuscript has been able to visually represent the magnitude of the migration capacity of the cells.

(13) Processed images are provided in the article for clearer image comparisons. However, all data were analysed by normalising the original images with Image J software. Raw images of the Western blotting assay have been provided in the supporting information files.

(14) Manuscripts have been fully checked and corrected for spelling errors. The language of the manuscript has been revised and corrected by International Science Editing Ltd.

(15) All gene expressions in the RT-PCT in the manuscript were revised to italics.

(16) The acronyms in the abstract have been replaced with the relative extended names+abbreviations.

(17) Reviewer 2 suggested the use of the 3D spheroid invasion assay to provide stronger supporting evidence for the study, which is a good comment. In further studies, we will adopt the above suggestions in order to further improve the quality of the study.

(18) Glycosylation has a significant impact on the physicochemical properties and biological functions of proteins, including protein stability, folding state, and interactions with other molecules.

(19) As suggested by reviewer 2, the glycosylation of CD147 is interpreted as follows: CD147 is a transmembrane glycoprotein with a molecular weight of approximately 43-66 kDa. Highly expressed in a variety of tumour cells and tissues. Functional studies have revealed that CD147 promotes tumour cell infiltration and metastasis by inducing peripheral fibroblasts and tumour cells to produce a variety of matrix metalloproteinases (MMPs), including MMP-1, MMP-2 and MMP-9, which degrade the extracellular matrix. Highly glycosylated modifications are one of the important features of CD147.

---

## [Decision Letter · Decision Letter 1]

4 Oct 2024

PONE-D-24-13072R1Monocarboxylate transporter dependent mechanism is involved in proliferation, migration, and invasion of human glioblastoma cell lines via activation of PI3K/Akt signalingpathwayPLOS ONE

Dear Dr. Gao,

Thank you for submitting your manuscript to PLOS ONE. After careful consideration, we feel that it has merit but does not fully meet PLOS ONE’s publication criteria as it currently stands. Therefore, we invite you to submit a revised version of the manuscript that addresses the points raised during the review process. **- Please define MCT, CD147, PBKVARt, VEGF, and siRNA acronyms at their first mention in both the abstract and the introduction**.

- Line 73: Correct "lacate" to "lactate".

**- British or American spelling (e.g., “tumor” vs. “tumour”) should be selected and used consistently throughout the manuscript.** **-Please better address reviewer's 2 questions.**

We look forward to receiving your revised manuscript.

Kind regards,

Nicola Amodio, PhD

Academic Editor

PLOS ONE

Reviewers' comments:

Reviewer's Responses to Questions

**Comments to the Author**

1. If the authors have adequately addressed your comments raised in a previous round of review and you feel that this manuscript is now acceptable for publication, you may indicate that here to bypass the “Comments to the Author” section, enter your conflict of interest statement in the “Confidential to Editor” section, and submit your "Accept" recommendation.

Reviewer #1: (No Response)

Reviewer #2: All comments have been addressed

Reviewer #3: (No Response)

2. Is the manuscript technically sound, and do the data support the conclusions?

Reviewer #1: Partly

Reviewer #2: Yes

Reviewer #3: Partly

3. Has the statistical analysis been performed appropriately and rigorously? 

Reviewer #1: Yes

Reviewer #2: Yes

Reviewer #3: Yes

4. Have the authors made all data underlying the findings in their manuscript fully available?

Reviewer #1: Yes

Reviewer #2: Yes

Reviewer #3: Yes

5. Is the manuscript presented in an intelligible fashion and written in standard English?

Reviewer #1: No

Reviewer #2: Yes

Reviewer #3: Yes

6. Review Comments to the Author

**Reviewer #1: **I appreciate the authors' efforts to improve their results and the quality of the manuscript. However, they have not addressed most, if not all, of the functional issues raised. In my opinion, the results presented are insufficient to support the findings claimed by the authors.

**Reviewer #2:** Since authors have performed all the changes requested by the reviewers, the manuscript is now acceptable for publication.

**Reviewer #3:** In this study. Dr. Gao and colleagues aim to clarify the mechanisms underlying MCT4/CD147 role in Glioblastoma progression and aggressiveness in human glioblastoma in vitro models. The manuscript was already reviewed and modified accordingly but I might suggest something else to revise before publication.

- Please define MCT, CD147, PBKVARt, VEGF, and siRNA acronyms at their first mention in both the abstract and the introduction.

- Line 73: Correct "lacate" to "lactate".

- British or American spelling (e.g., “tumor” vs. “tumour”) should be selected and used consistently throughout the manuscript.

- Please consider including more specific information in the transwell method section about the statistical methods used to validate experimental results. Moreover, the added part after revision seems to be taken directly from a protocol and not contextualized.

- Ensure consistency when referring to “siRNA interference” or “siRNA knockdown.”

- The revised manuscript includes raw Western blot images, as requested. Make sure that figure legends are adequately described for example by adding how images were quantified using ImageJ (mentioned in the Results but could be added in the captions as well).

- Consider adding some recent (last 2 years) literature on MCT4/CD147 role in glioblastoma as references.

- Please also state the rationale for using ANOVA with post hoc tests, and if it is well justified for your specific requirements and data.

- The revised discussion has improved but can still be more cautious in the mechanistic conclusions you made. For example, when discussing MCT/ CD147 and Akt signaling, consider removing the assumption with something like suggestion or hypothesis.

Overall, my suggestion to the editor is Minor Revisions.

7. PLOS authors have the option to publish the peer review history of their article (what does this mean?). If published, this will include your full peer review and any attached files.

Reviewer #1: No

Reviewer #2: **Yes: **Anna Martina Battaglia

Reviewer #3: No

---

## [Author Response · Author response to Decision Letter 1]

14 Oct 2024

(1) All the acronyms, such as MCT, CD147, PBKVARt, VEGF and siRNA had been defined at their first mention in both the abstract and the introduction.

(2) Line 70, the word “lacate” has been amended to “lactate”.

(3) Throughout the text of the manuscript, the word tumour is standardised to the British expression, ‘tumor’, but in the references, the American expression, ‘tumour’, is still used in some of the literature (reference 20 and 22).

(4) Closely in context, a more detailed description of the Transwell invasion assay method is provided in the Materials and Methods section, specifically describing the use of Image J software for counting migrating and invaded cells.

(5) To ensure consistency, line 296 and 362, the word “knockdown” has been amended to “interference”, line 359, the word “knockdown” has been amended to “silencing”.

(6) The method of immunoblotting image analysis with Image J software is described in detail in the Materials and Methods section.

(7) Three updated references on MCT4/CD147 role in glioblastoma have been to the Introduction, Materials and Methods, and Discussion sections, respectively. (Fernandez GB, et al. Cells, 2022. doi: 10.3390/cells11213467; Longhitano L et al. Front Oncol. 2022. doi: 10.3389/fonc.2022.871798. eCollection 2022; Kubelt C et al. Int J Mol Sci. 2020. doi: 10.3390/ijms21176254.)Reference numbers have been revised accordingly.

(8) Numerical data that are normally distributed can be analyzed with parametric tests, that is, tests which are based on the parameters that define a normal distribution curve. The one-way analysis of variance (ANOVA) is employed to compare the means of three or more independent data sets that are normally distributed. The quantitative data within the group in this study conformed to normal distribution, so one-way ANOVA was used for statistical treatment. Depending on results of homogeneity of Variance Test, the final choice of either Tamhane or Bonferroni correction was made for the between-group comparison results. The above is explained in the statistical analysis section.

(9) In the final part of the discussion section, expressions such as “suggest” and “hypothesis” are used, making it more cautious to draw research conclusions.

---

## [Editor Report · Decision Letter 2]

16 Oct 2024

Monocarboxylate transporter dependent mechanism is involved in proliferation, migration, and invasion of human glioblastoma cell lines via activation of PI3K/Akt signalingpathway

PONE-D-24-13072R2

Dear Dr. Gao,

We’re pleased to inform you that your manuscript has been judged scientifically suitable for publication and will be formally accepted for publication once it meets all outstanding technical requirements.

Kind regards,

Nicola Amodio, PhD

Academic Editor

PLOS ONE
---

## [Editor Report · Acceptance letter]

18 Oct 2024

PONE-D-24-13072R2 

PLOS ONE

Dear Dr. Gao, 

I'm pleased to inform you that your manuscript has been deemed suitable for publication in PLOS ONE. Congratulations! Your manuscript is now being handed over to our production team.

Kind regards, 

on behalf of

Dr. Nicola Amodio 

Academic Editor

PLOS ONE